# Differences in Health Behavior Profiles of Adolescents in Urban and Rural Areas in a Korean City

**DOI:** 10.3390/healthcare9030282

**Published:** 2021-03-04

**Authors:** Myungah Chae, Kihye Han

**Affiliations:** College of Nursing, Chung-Ang University, Seoul 06974, Korea; chae0326@naver.com

**Keywords:** adolescence, health behaviors, latent class analysis, region, Korea

## Abstract

Through a latent class analysis approach, we can classify individuals and identify subgroups according to health behavior patterns, and find evidence for the development of customized intervention programs to target high-risk groups. Our study aimed to explore differences in latent classes of health behaviors in adolescents by region (urban vs. rural areas) in a Korean city. This cross-sectional secondary analysis utilized data collected from all first graders’ student health checkups in middle school and high school in a city of the largest island in Korea in 2016 (n = 1807). Health behavior indicators included both healthy (consuming breakfast regularly, consuming vegetables daily, consuming milk daily, consuming fast food on a limited basis, engaging in vigorous physical activities, brushing teeth, and practicing hand hygiene) and unhealthy (drinking, smoking, and overusing the internet) behaviors. Nutritional and diet behaviors were important factors for classifying healthy and unhealthy adolescents in both regions. Approximately 11% of rural students belonged to the risky group, which was characterized by a high level of drinking alcohol and smoking. These results suggest that when developing health policies for adolescents, customized policy-making and education based on the targeted groups’ behavioral patterns could be more effective than a uniform approach.

## 1. Introduction

Adolescence is a turning point in life, accompanied by sudden physical, mental, and emotional changes. Adolescents’ health behaviors are influenced by various factors, such as ethnicity, culture, and social environment, and they may engage in some risky health behaviors [1]. Managing adolescent health and behavior is essential as they significantly influence their future development of chronic diseases [2,3]. Health-promoting behaviors should be increased in adolescents, including adequate nutrition and physical activity levels, and they should be made aware of the dangers of risky behaviors to reduce negative health outcomes [4].

Many existing policies focus on individual-level health behaviors. As the relationship between health behaviors is not the same across individuals, person-oriented techniques have been used to identify subgroups [5]. Through a latent class analysis approach, we can classify individuals based on their scoring patterns. This analytic approach determines important classification factors and provides evidence for the development of customized interventions targeting high-risk groups and including focused education and health policies [6].

In the past, health disparity was blamed on individual health behaviors or lifestyles, but socioeconomic status is the most influential long-term factor contributing to health discrepancies [7]. People choose where to reside in part according to their socioeconomic status. This geographical community affects wellbeing and influences everyday life, making it a primary factor in adolescent health [8,9]. The social capital in the family, school, and community domains is a critical element for health maturation and a preventive factor in adolescent risk behaviors. Previous studies have reported that rural areas or small- and medium-sized cities lack healthy leisure and cultural facilities compared to large urban areas with rich cultural resources and more political and economic power [10,11,12]. Adolescents in rural areas are exposed earlier to more hazardous and risky behaviors, including smoking and drinking. Adolescents in large cities are less likely to engage in vigorous- or moderate-intensity physical exercise than those in rural or small- and medium-sized cities [2]. Comparing the health behavior patterns between adolescents in different communities is significant to health advocacy agencies and policymakers.

In Korea, adolescent health behaviors are periodically assessed through the Korea Youth Risk Behavior Web-based Survey (http://www.kdca.go.kr/yhs/ (accessed on 4 March 2021)) [13]. This study uses stratified proportional sampling to ensure representation of the entire country. The samples from a small region may not fully reflect regional characteristics. For example, adolescents on the largest island in South Korea comprise 2.3% of the sample, which is not large enough to be representative. Islands may be economically, culturally, and administratively isolated and be far from the mainland. Daily living activities may also be limited due to the land being narrow. According to previous studies conducted in different countries, the health status of adolescents on an island was distinct from that of adolescents on the mainland [14]. Research is lacking on adolescent health behaviors in the island areas of Korea. More studies with extensive sampling for generalizability are necessary to investigate these regional characteristics.

This study used data collected from first graders’ student health checkups in middle school and high school in a particular city on the largest island in Korea in 2016. We aimed to compare the adolescent health behavior patterns between urban and rural areas.

## 2. Materials and Methods

### 2.1. Study Design and Setting

This cross-sectional secondary analysis utilized data from all first-year students in middle and high schools in a city on the largest island (Jeju-do) in Korea (n = 1807).

### 2.2. Data Sources and Participants

Self-reported health behavior data were obtained from the health checkup information of 1807 students: first graders in middle school (7th graders, approximately 13 years old, n = 845) and first graders in high school (10th graders, approximately 16 years old, n = 962) in 2016. In Korea, the Office of Education conducts annual country-wide student health checkups as a part of the school health examination project [7]. The checkups include a comprehensive survey in first- and fourth-graders in elementary school, first graders in middle school, and first graders in high school, so each student receives a health checkup every three years. The survey consists of questions regarding illnesses in the recent year, symptoms experienced in the recent month, and health behaviors, such as diet, personal hygiene, physical exercise, safety, internet use, mental health problems, smoking, and drinking. Additionally, a physical examination is conducted, including blood tests, a chest X-ray, a urine test, and height, weight, and blood pressure measurements. These data are not publicly accessible and are not included in this study.

### 2.3. Measures

For healthy behaviors which positively influence health, we included the following items in this study: consuming a regular breakfast, consuming vegetables and milk daily, limiting fast food consumption, engaging in vigorous physical activities, brushing teeth regularly, and practicing good hand hygiene. We assessed the risky behaviors of drinking, smoking, and internet overuse, which all negatively affect health. All of the health behavior indexes in this study were measured as single items, with a (yes/no) option for each: having breakfast regularly, having fruits and vegetables daily, having fast food daily, having milk or dairy products daily, engaging in vigorous physical activity, brushing teeth more than twice a day, washing hands before having meals and when coming home after being out, drank alcohol in the past 30 days, smoked in the past 30 days, and used the internet or internet games for more than two hours a day.

Regions were categorized based on school location: urban areas (a large-sized area with more than 180,000 residents) vs. rural areas (small- and medium-sized areas with less than 50,000 residents).

### 2.4. Data Analysis

To identify the health behavior profiles of adolescents, we stratified data by urban and rural areas and separately performed latent class analysis using Mplus version 8 [15]. Latent class analysis (LCA) is a person-centered approach to determine heterogeneous groups of cases (i.e., clusters) based on the probabilities that individuals report regarding each health behavior indicator. Based on an assumption of the observed indexes’ mutual independence, latent class analyses were repeated by increasing the number of classes from one. The number of latent classes of the best model was determined based on the combination of model fit indexes. For the Akaike information criterion (AIC) and Bayesian information criterion (BIC), indicating goodness of fit and parsimony, the lower the value, the better the model. The Lo–Mendell–Rubin likelihood ratio test (LMR-LRT) compares the model fit improvement between the k and k-1 classes. Values < 0.05 represent a significant improvement for an increased number of classes [16]. The entropy index indicates the classification accuracy. It ranges between zero and one, with a value closer to one indicating a more accurate classification [17]. For optimal model selection, statistical criteria and a comprehensive review of the models and latent classes are essential to understand the model’s simplicity and interpretability and the latent classes distribution [18]. To easily interpret the findings, all of the health behavior indexes were presented in the table and figures as higher probabilities indicating poorer behavior practice. 

## 3. Results

Table 1 presents the model fit statistics for LCA models of two- to five-class solutions among students in urban and rural areas. The BIC value of the two latent class models for students in urban areas (left column in Table 1) was smaller than those for the three- to five latent class models. The LMR-LRT was significant in the two and five latent class models. Although the five-class model had the highest entropy value, class four had a class count of <5%. This suggests that the best overall model fit was the parsimonious identification of the two latent classes in urban students. 

Figure 1a provides a graphical representation of the two latent classes of students in urban areas. The x-axis represents the health behaviors and the y-axis shows the probability of having poor health behaviors within each class. We labeled each class based on health behavior probabilities (λ) and identified two health behavior profiles for the students in urban areas (left column in Table 2). The first profile was the healthy group (Figure 1a, dotted line), comprising about 53% of the urban-area students. The healthy group had high probabilities of health-promoting behaviors, such as regularly consuming breakfast (λ = 0.91), vegetables (λ = 0.93), and milk (λ = 0.71), and engaging in physical exercise (λ = 0.61). This group had low probabilities of health risk behaviors such as drinking (λ = 0.01) and smoking (λ = 0.01). The second profile was the unhealthy group (Figure 1a, solid line), comprising about 47% of those in urban areas. Relatively few students in the unhealthy group reported practicing health-promoting behaviors, such as milk consumption (λ = 0.28) and engaging in physical exercise (λ = 0.32). Members in this group reported low probabilities of risky behaviors such as drinking (λ = 0.02) and smoking (λ = 0.02).

For students in rural areas, the three latent class model showed the lowest AIC and BIC values for parsimonious identification, although the highest entropy value was found in the two-class model (right column in Table 1). The LMR-LRT was significant in the two- and three-class models, so the three latent class model was chosen. The first profile was the healthy group (Figure 1b, dotted line), comprising about 41% of the rural-area students. This healthy group showed relatively high probabilities for health-promoting behaviors such as consuming breakfast (λ = 0.72), vegetables (λ = 0.86), and milk (λ = 0.63) and engaging in physical exercise (λ = 0.59) (right column in Table 2). In contrast, they had low probabilities for risky behaviors such as drinking (λ = 0.04) and smoking (λ = 0.04). The second profile was the unhealthy group (Figure 1b, solid line), comprising about 47% of those in rural areas. Members in this group reported low probabilities for health-promoting behaviors such as consuming breakfast (λ = 0.36), vegetables (λ = 0.21), and milk (λ = 0.22) and engaging in physical exercise (λ = 0.31). They showed low probabilities of risky behaviors such as drinking (λ = 0.05) and smoking (λ = 0.04). The third profile was the risky group (Figure 1b, gray line) comprising about 11% of rural adolescents. Probability values in the risky group were very high for risky behaviors such as drinking (λ = 0.84) and smoking (λ = 1.00).

## 4. Discussion

This investigation is highly generalizable for the study area as it utilized a comprehensive set of data from all first graders in the middle and high schools in a city on the largest island in Korea to classify health behavior characteristics. The results indicate that essential health behaviors for the subgroup classification differ by geographical region. This suggests that when developing health policies targeting adolescents, customized policy-making and education based on group characteristics could be more effective than uniformly approaching all students. Latent class analysis is useful for identifying subgroups for future interventions.

We identified two classes (healthy and unhealthy) for the urban area students and three classes (healthy, unhealthy, and risky) for the rural area students using latent class modeling to examine the clustering effects of health behavior variables among adolescents. We included ten health behavior variables, so two or three classes in the health behavior profiles were generated [19]. As there has been limited research about health behavior patterns among Korean adolescents, our findings could not be compared and/or explained based on previous studies. In U.S. adolescent studies, health risk behaviors were classified into two and four groups among 7th and 9th graders [20], whereas four groups of health behaviors were extracted among 9th to 12th graders [21]. More heterogeneous subgroups tend to be found in higher graders [20] and when a variety of health behaviors are included, such as self-injury and suicide risk behaviors [22]. Our study included young adolescents expected to have few health problems and did not use other health variables. Previous studies extracted more than five classes based on various chronic health conditions and health service accessibility issues for elderly populations [23,24] and diverse intentions toward health, like influenza vaccinations and doctor visits, for adult populations [25]. More diverse profiles could be created by including other relevant health items, such as weight and stress management. Furthermore, rare but possible juvenile delinquency groups could be captured by statistical modeling using high-risk sexual behaviors, recreational substance use, etc.

The first two classes within each region (i.e., healthy and unhealthy groups) were differentiated by the level of nutrition-related behaviors, suggesting that regular consumption of breakfast, vegetables, and/or milk is an essential component of classifying health behavior patterns. This indicates a need for education on proper dietary habits. Nutritional enhancement strategies are necessary, such as providing breakfast at schools or school meals containing nutritious fresh foods, including salads and milk.

It should be noted that for the health behavior classifications in rural areas, the risky group comprised 11% of the students. Students in this group showed high levels of consuming alcohol and smoking. Compared to the students in urban areas, those in rural areas were more frequently offered their first cigarette by their father or other adults [26]. They also reported their schools, homes, or friends’ houses as places they frequently smoked, rather than in locations downtown. This may be due to the relatively weak social norms or regulations regarding adolescent drinking and smoking in rural regions, which make it easier to engage in these risky behaviors [2]. This suggests that students in rural areas were more frequently exposed to smoking and drinking and that community-based educational programs for parents and adults are necessary.

Our study applied the latent class analysis technique to investigate adolescent health behaviors and compared health behavior profiles by region. There are some limitations to this study. First, secondary data analysis did not permit the use of additional factors, such as parent information, the standard of living, and lifestyle details, which were not measured in the original project. Factors influencing adolescents’ health include home environmental factors, such as family structure, household income, and the parents’ education levels [27]. Parental socioeconomic status, dysfunctional family environments, and healthcare disparities among rural and urban communities also indirectly affect children’s eating habits and health management techniques [28]. Second, because all data were self-reported, adolescents’ responses might be affected by recall bias or denial/social desirability, especially regarding negative behaviors. The use of objective health behaviors or outcomes (e.g., body mass index) data could provide more reliable results. Finally, the school administrative districts were used to classify the region into urban and rural areas to compare geographical differences in health behaviors. As the residential location was not included, different lifestyles might not be fully reflected in this study.

## 5. Conclusions

This secondary analysis used comprehensive health checkup data of all first-grade students in middle and high schools in a city and compared health promotion and risk behavior patterns by geography. Important health behaviors in classifying the patterns differed according to region, indicating that the development and implementation of adolescent health policies should be customized by population characteristics. For adolescents in both regional areas, nutritional and diet behaviors were important in classifying healthy and unhealthy groups. Approximately 11% of rural students belonged to the risk group, which was characterized by high levels of alcohol consumption and smoking.

For health-promoting interventions, contextual community elements should be considered. A comprehensive school health program in the United States reported a significant improvement in students’ health (e.g., obesity rates). The program offers students healthy school meals and extended opportunities for physical activities [29,30,31]. In Korea, there is little research on ecological interventions to resolve adolescent health disparities. We suggest a collaboration between education authorities and the local community for reducing adolescent health disparities in different regions. Adolescents’ early initiation of risky behaviors should be considered to successfully develop community-based health-promoting programs in rural areas. Projects to promote adolescent health should be initiated during middle school. The age of adolescent smoking and drinking onset in Korea has decreased recently [32,33]. Those who start smoking and alcohol drinking before the age of 19 years find it more difficult to quit than those who start after the age of 20. This suggests that adolescence is a crucial period for risk behavior interventions [34,35,36].

To increase health promotion engagement of adolescents, school nurses should provide educations on self-care management strategies for healthy dietary habits in the fundamental curriculum of middle and high schools. Community health nurses should include both adults and adolescents in educational sessions regarding the long-term harm of early exposure to risky behaviors. To decrease risky health behaviors among adolescents, we suggest that mandatory education regarding risky health behaviors be included in schools at an early age and provided.

Our study elucidates a pattern classification of adolescent health behaviors and identifies significant health behavior indicators within different geographical groups. Future studies should include family factors that might affect the pattern classification of adolescent health behaviors, such as family structure and family environment.

## Figures and Tables

**Figure 1 healthcare-09-00282-f001:**
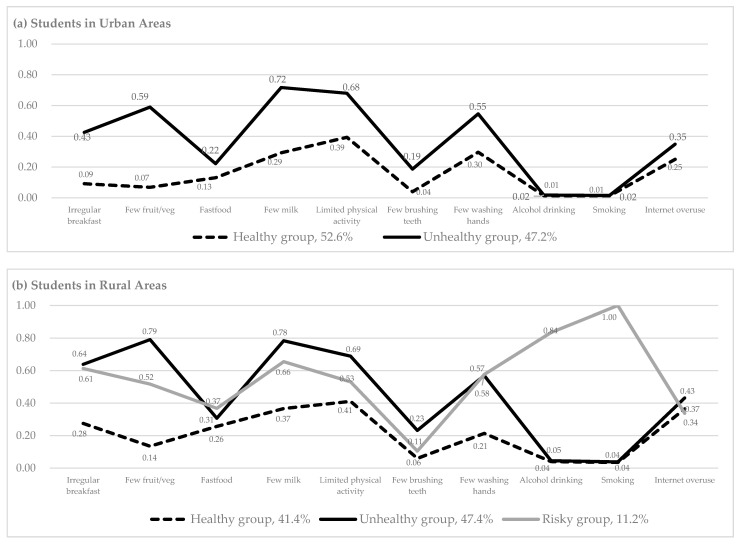
Health behavior characteristics for each latent class. (**a**) Students in urban areas, (**b**) students in rural areas.

**Table 1 healthcare-09-00282-t001:** Model fit statistics for two- to five-class models, students in urban vs. rural areas.

		Students in Urban Areas				Students in Rural Areas			
Model Fit Indexes		2	3	4	5	2	3	4	5
AIC		10,847.64	10,826.976	10,817.361	10,822.263	7279.136	7147.319	7137.358	7126.431
BIC		10,953.601	10,988.441	11,034.33	11,094.735	7373.153	7290.582	7329.868	7368.188
LMR-LRT		0	0.1366	0.2312	0.0236	0	0	0.0777	0.197
Entropy		0.486	0.487	0.587	0.67	0.856	0.699	0.726	0.668
Class count	1	604 (52.6%)	532 (46.3%)	375 (32.7%)	443 (38.6%)	540 (83.1%)	269 (41.4%)	56 (8.6%)	219 (34.0%)
	2	544 (47.4%)	382 (33.3%)	20 (1.7%)	210 (18.3%)	110 (16.9%)	308 (47.4%)	248 (38.2%)	186 (28.6%)
	3		234 (20.4%)	538 (46.9%)	222 (19.3%)		73 (11.2%)	299 (46.0%)	47 (7.2%)
	4			215 (18.7%)	15 (1.3%)			47 (7.2%)	55 (8.5%)
	5				258 (22.5%)				143 (22.0%)

AIC, Akaike information criterion; BIC, Bayesian information criterion; LMR-LRT, Lo–Mendell–Rubin likelihood ratio test.

**Table 2 healthcare-09-00282-t002:** Probabilities of health behavior indicators by class membership among students in urban and rural areas.

		Students in Urban Areas (n = 1152, 63.8%)			Students in Rural Areas (n = 655, 36.2%)			
			Healthy Group	Unhealthy Group		Healthy Group	Unhealthy Group	Risky Group
Health Behavior Indicators		Overall	(52.6%)	(47.4%)	Overall	(41.4%)	(47.4%)	(11.2%)
Having breakfast regularly	no	0.26	0.09	0.43	0.48	0.28	0.64	0.61
	yes	0.74	0.91	0.57	0.52	0.73	0.36	0.39
Having fruits and vegetables everyday	no	0.33	0.07	0.59	0.49	0.14	0.79	0.52
	yes	0.68	0.93	0.41	0.51	0.87	0.21	0.48
Having fast food everyday	yes	0.18	0.13	0.22	0.29	0.26	0.31	0.37
	no	0.82	0.87	0.78	0.71	0.74	0.69	0.63
Having milk or dairy products everyday	no	0.50	0.29	0.72	0.60	0.37	0.78	0.66
	yes	0.50	0.71	0.28	0.40	0.63	0.22	0.35
Engaging in vigorous physical activity	no	0.54	0.39	0.68	0.56	0.41	0.69	0.53
	yes	0.47	0.61	0.32	0.45	0.59	0.31	0.47
Brushing teeth more than twice a day	no	0.11	0.04	0.19	0.15	0.06	0.23	0.11
	yes	0.89	0.96	0.81	0.86	0.94	0.77	0.90
Washing hands prior to having meals or when coming home after being out	no	0.42	0.30	0.55	0.42	0.21	0.57	0.58
	yes	0.58	0.70	0.45	0.58	0.79	0.43	0.42
Drank alcohol in the past 30 days	yes	0.02	0.01	0.02	0.13	0.04	0.05	0.84
	no	0.98	0.99	0.98	0.87	0.96	0.96	0.17
Smoked in the past 30 days	yes	0.01	0.01	0.02	0.14	0.04	0.04	1.00
	no	0.99	0.99	0.99	0.87	0.97	0.96	0.00
Use internet or internet games more than two hours a day	yes	0.30	0.25	0.35	0.39	0.37	0.43	0.34
	no	0.70	0.75	0.65	0.61	0.64	0.57	0.66

## Data Availability

Publicly available datasets were analyzed in this study. This data can be found at: [http://www.schoolhealth.kr/web/srs/selectPrivacyAgree.do?sMenuId=0100008800 (accessed on 4 March 2021)].

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
