# Peer review of "Differences in Health Behavior Profiles of Adolescents in Urban and Rural Areas in a Korean City"

_healthcare, 2021, doi:10.3390/healthcare9030282_

Round 1

Reviewer 1 Report

Thanks for the opportunity to review the article named “Differences in health behavior profiles of adolescents in urban 2 and rural areas in a Korean city” by Myungah Chae and Kihye Han. This study focuses on identifying adolescents with unhealthy behavior patterns for effective targeted intervention. 

The study results should be analyzed with due consideration about the reliability of self-reported health behavior patterns, especially when it comes to reporting negative behavior patterns like smoking and drinking behaviors. It might have some component of self-reporting bias which should be reported in the limitation. 
-       Including more information like BMI might provide objective data to reduce self-reporting bias. DOI:https://doi.org/10.1016/j.amjmed.2020.04.018
-       The authors stated the impact of socioeconomic factors and geographical community in developing adolescent’s health behaviors. The other factors that could potentially confound the study results are dysfunctional family environment, healthcare disparity among rural and urban communities.
-       The authors discussed nutrition, exercise, personal hygiene, smoking, and drinking for categorizing healthy versus unhealthy groups. The other behavior patterns like high-risk sexual behavior, recreational substance use, illegal behavior, risky driving are not addressed. 
-       Please include the copy of the questionnaire “The Korea Youth Risk Behavior Web-based Survey (KYRBS)” used
-       Gender classification of study participants as boys and girls will help identify which sub-group is more prone to unhealthy behaviors and for targetted intervention

Author Response

Response to Reviewer 1 Comments

Comments and Suggestions for Authors

Thanks for the opportunity to review the article named “Differences in health behavior profiles of adolescents in urban 2 and rural areas in a Korean city” by Myungah Chae and Kihye Han. This study focuses on identifying adolescents with unhealthy behavior patterns for effective targeted intervention.

  1. The study results should be analyzed with due consideration about the reliability of self-reported health behavior patterns, especially when it comes to reporting negative behavior patterns like smoking and drinking behaviors. It might have some component of self-reporting bias which should be reported in the limitation. Including more information like BMI might provide objective data to reduce self-reporting bias. DOI: https://doi.org/10.1016/j.amjmed.2020.04.018

Thank you for pointing this out. We have added information regarding potential self-reporting bias in the limitation section as suggested (lines 227-230).

  1. The authors stated the impact of socioeconomic factors and geographical community in developing adolescent’s health behaviors. The other factors that could potentially confound the study results are dysfunctional family environment, healthcare disparity among rural and urban communities.

Thank you for your suggestion. We have included other factors that could affect the study results in the limitation section (lines 225-227).

  1. The authors discussed nutrition, exercise, personal hygiene, smoking, and drinking for categorizing healthy versus unhealthy groups. The other behavior patterns like high-risk sexual behavior, recreational substance use, illegal behavior, risky driving are not addressed.

Thank you for your suggestion. We have mentioned the potential use of high-risk health behaviors in the Discussion section (lines 200-202).

  1. Please include the copy of the questionnaire “The Korea Youth Risk Behavior Web-based Survey (KYRBS)” used

We have added the URL (http://www.kdca.go.kr/yhs/), which directly links to the Korea Youth Risk Behavior Web-based Survey (KYRBS) questionnaire (line 55).

  1. Gender classification of study participants as boys and girls will help identify which sub-group is more prone to unhealthy behaviors and for targetted intervention

Thank you for your suggestions. As a separate study aim, our research team compared health behavior patterns by gender and developed a manuscript that is now under review at the International Journal of Environmental Research and Public Health (manuscript ID: IJERPH- 1118171).

Reviewer 2 Report

The manuscript with ID: healthcare-1129766, and title: Differences in health behaviour profiles of adolescents in urban and rural areas in a Korean city is an interesting document and I think it should be published with slight modifications.

Line 72, it would be adequate to give the name of the largest island in Korea.

Line 77, as the education system is not the same in all the countries it would help to understand the study to add the average age of students in the two groups here considered.

I think that the differences in healthy habits between urban areas and rural areas are very interesting, but I miss some information about the differences in the way of living among rural areas and urban areas. In rural areas, the population have lower student levels. Are they farmers? -…Income differences,….Please add some data if it is possible.

Author Response

Response to Reviewer 2 Comments

Comments and Suggestions for Authors

The manuscript with ID: healthcare-1129766, and title: Differences in health behaviour profiles of adolescents in urban and rural areas in a Korean city is an interesting document and I think it should be published with slight modifications.

  1. Line 72, it would be adequate to give the name of the largest island in Korea.

We have provided the name of the largest island, as suggested (line 72).

  1. Line 77, as the education system is not the same in all the countries it would help to understand the study to add the average age of students in the two groups here considered.

We have provided the ages of the students in the two groups, as suggested (lines 76-78).

  1. I think that the differences in healthy habits between urban areas and rural areas are very interesting, but I miss some information about the differences in the way of living among rural areas and urban areas. In rural areas, the population have lower student levels. Are they farmers? -…Income differences,….Please add some data if it is possible.

Unfortunately, our secondary data analysis study could not include environmental or parental information as this was not measured in the original project. We included this information in the limitation section (lines 221-227).

Round 2

Reviewer 1 Report

The authors have addressed all my concerns and mentioned them in their limitations. 

No further comments.

Author Response

Response to Reviewer 1 Comments

Comments and Suggestions for Authors

The authors have addressed all my concerns and mentioned them in their limitations. No further comments.

Thank you for your time and valuable comments to improve our manuscript.
